# Design and Measurement of a Dual FBG High-Precision Shape Sensor for Wing Shape Reconstruction

**DOI:** 10.3390/s22010168

**Published:** 2021-12-28

**Authors:** Huifeng Wu, Lei Liang, Hui Wang, Shu Dai, Qiwei Xu, Rui Dong

**Affiliations:** 1National Engineering Laboratory for Fiber Optic Sensing Technology, Wuhan University of Technology, Wuhan 430070, China; huizifeng_829@163.com (H.W.); wanghui1989@whut.edu.cn (H.W.); ds0567@whut.edu.cn (S.D.); xxdsgt2021@163.com (Q.X.); 2School of Electronic Information and Automation, Guilin University of Aerospace Technology, Guilin 541004, China; 3Engineering Comprehensive Training Center, Guilin University of Aerospace Technology, Guilin 541004, China; dongrui87856@163.com

**Keywords:** fiber Bragg grating, curvature, torque, shape reconstruction

## Abstract

FBG shape sensors based on soft substrates are currently one of the research focuses of wing shape reconstruction, where soft substrates and torque are two important factors affecting the performance of shape sensors, but the related analysis is not common. A high-precision soft substrates shape sensor based on dual FBGs is designed. First, the FBG soft substrate shape sensor model is established to optimize the sensor size parameters and get the optimal solution. The two FBG cross-laying method is adopted to effectively reduce the influence of torque, the crossover angle between the FBGs is 2α, and α = 30° is selected as the most sensitive angle to the torquer response. Second, the calibration test platform of this shape sensor is built to obtain the linear relationship among the FBG wavelength drift and curvature, rotation radian loaded vertical force and torque. Finally, by using the test specimen shape reconstruction test, it is verified that this shape sensor can improve the shape reconstruction accuracy, and that its reconstruction error is 6.13%, which greatly improves the fit of shape reconstruction. The research results show that the dual FBG high-precision shape sensor successfully achieves high accuracy and reliability in shape reconstruction.

## 1. Introduction

Fiber grating shape sensing technology has been a new research direction in the field of fiber grating sensing in recent years, especially in three-dimensional shape reconstruction, and it is a widely studied and researched method that uses FBG shape sensors to obtain shape change information parameters [1], for example, in position tracing and positioning of probes, soft manipulators, endoscopes, etc., in the medical field [2,3]; deformation monitoring of key structures in the aerospace field, such as deformation monitoring of airplane skin, wings, etc. [4,5,6]; real-time monitoring of the motion status and spatial position of mechanical arms embedded in robot links and robot arms of continuum robots in industrial fields [7,8,9]; realization of joint positioning and posture monitoring during human movement by flexible smart wearable devices [10,11]; and the structural health monitoring of urban civil buildings [12] and bridges [13]. 

After MIT and NASA jointly designed a new digital deformable wing with polyimide film skin and composite lattice honeycomb structure, the development and application of FBG shape sensors are mainly concentrated in two types: Shape sensor based on a single core fiber grating, and shape sensor based on a multicore fiber.

A multicore fiber shape sensor is suitable for extremely large curvatures (bending radii up to 5 mm). As early as 2012, the US Luna set a multicore fiber shape sensor (total length of 30 m) on the surface of flexible structures of approximately 10 m, obtained the space deformation of the measured surface by reconstructing the multicore fiber space position, and had found that the reconstruction error was within 1.5% [5]. In recent years, distributed optical fiber shape sensors have developed rapidly. Chen, X.Y et al. established a distributed Brillouin optical time domain analysis based on a seven-core MCF to verify the linear relationship between fiber curvature and the Brillouin frequency shift of the outer core in the MCF and realized a curvature with a spatial resolution of 20 cm using a 1 km fiber length [14]. According to the temperature-sensitive characteristics of optical fibers, Zhao, Z et al. designed a high-performance temperature-insensitive shape sensor based on a differential pulse pair system, whose spatial resolution reached 10 cm, and its sensitivity was greatly improved [15]. Ba, D.X et al. proposed a high-sensitivity distributed shape sensor based on a seven-core optical fiber and phase sensitive optical time domain reflectometry, which can achieve high strain sensitivity as low as 0.3 με on a 24 m-long MCF with a spatial resolution of 10 cm [16]. 

Some of the inevitable problems of using multicore optical fibers as shape sensors are twisting, eliminating the measurement error caused by distortion and improving the accuracy of shape measurement and reconstruction, which have become the focus of research for many scholars in recent years. Some scholars have proposed solutions for spiral multicore fibers [17], but the high fabrication cost and immature fabrication technology of spiral multicore fibers prevent their large-scale application.

In comparison, the moldability of a single-core fiber grating shape sensor is better, especially the combined sensing mode of a soft substrate material and FBG, which is more widely used [18]. Wang et al. proposed a sensing network and method-based FBG sensor for 3D shape detection of flexible robotic arms and proved high detection accuracy for complex 3D shapes by simulation [19]. Zhi, G.H et al. designed a new intensity modulated fiber optic sensor with an enhanced sensitivity region to measure curvature and proposed a wind speed monitoring system based on the fiber optic shape sensor, which monitored the frequency of the fiber optic sensor period signal to measure wind speed. The experimental results show that the measurement error and repeatability of the wind speed monitoring system are within ±0.65 m/s and ±2.9%, respectively [20]. Abro, A. Z et al. [21,22] developed and designed a smart sensing garment SSG and a FBG smart belt. The FBG was embedded in special silicone to make a shape sensor that was mounted on the palm, wrist and elbow joints to measure changes in body posture. The sensitivity of the FBG smart belt was between 0.018% and 0.021% for male subjects at speeds of 2 and 3 km/h, respectively. He, Y.L et al. [23,24,25] used silica gel and polyimide with FBG to design two different types of shape sensors; one was a soft actuator based on silica gel to reconstruct the 3D shape of the soft actuator in different bending states, and the other was to embed FBG into polyimide to make a morphing wing smart skin. The 3D shape reconstruction of the polyimide skin of the wing was completed. The test results show that the maximum error of FBG measurement is less than 5%. Wang, Q.L et al. [26] proposed a bending curvature sensor for a soft surgical manipulator, which buried a Bragg grating into polyimide to form a polyimide sensing layer and integrated the sensing layer into a soft manipulator model to achieve bending curvature monitoring of the soft surgical manipulator. The maximum error between the measured value and the actual bending curvature was less than 2.1%. Arnaldo G et al. [27] presented a 3D displacement sensor based on a single fiber Bragg grating in CYTOP fiber, to obtain the influence of each displacement condition, namely, axial strain, torsion and bending on the FBG reflection spectrum, and had relative errors below 5.5%, but the fiber Bragg grating in CYTOP was prone to breakage and not suitable for harsh environments.

For single-core fiber grating shape sensors, how to eliminate the measurement error caused by distortion to improve the accuracy of shape reconstruction was not considered in the previous literature. In addition, there is little literature on how to eliminate the influence of the properties of soft substrate materials and volume size on the performance of shape sensors to improve the measurement accuracy of sensors, and only a few papers have analyzed the effect of the depth of FBG preburial in soft substrates [28,29,30]. 

In this paper, we choose the more widely used single-core FBG and soft substrate combination mode to design a new FBG shape sensor, analyze in detail the influence of the length, thickness and width of the soft substrate on the accuracy of the shape sensor and choose the most reasonable soft substrate size. At the same time, considering the irresistible factor of torque in the use of FBG shape sensors, we adopt the cross double-grid method to measure the FBG center wavelength drift caused by torque. By a calibration test, the linear relationship between FBG wavelength drift and rotation radian loaded torque is obtained. The shape sensor designed in this paper can structurally improve the measurement accuracy.

## 2. FBG Sensing Principle and Structure Analysis

### 2.1. Sensory Principle of FBG

From Hooke’s law, we know that after a uniform elastic solid material is stressed, there is a linear relationship between stress ε and strain (unit deformation) in the material, as shown in Equation (1) [31].
(1)$$\varepsilon =\frac{\Delta l}{l}$$

A microarc segment of the elastic material is intercepted, in which length L and thickness H are chosen, and the location of the preburied FBG is X from the upper plane, as shown in Figure 1. When the elastic material is subjected to axial stress, two cross-sections of the microarc segment relatively rotate by a microsmall angle *θ*, as shown in Figure 1b. The length of the neutral plane does not change, the length of the upper plane is stretched by Δl, the length of the under plane is shortened by Δl and the length of the plane on which the FBG is located is stretched by Δx, where Δx<Δl and R is the radius of curvature.

When the length of the microarc segment is small enough, sinθ=θ
(2)upper plane:  L+ ΔL=(R+H)θ
(3)$$\mathrm{neutral}\;\mathrm{plane}:\;\;L=\left(R+\frac{1}{2}H\right)\theta$$
(4)under plane:  −ΔL=Rθ

From Equations (2) and (4), the following is obtained: (5)2ΔL=Hθ 

From Equations (3) and (5), the following is obtained:(6)$$\frac{\Delta L}{L}=\frac{\frac{1}{2}H\theta }{\left(R+\frac{1}{2}H\right)\theta }=\frac{\frac{1}{2}H}{R+\frac{H}{2}}\;$$

For a standard single--method optical fiber, the relationship between the center wavelength drift  Δλ and the center wavelength λ and the changes in temperature ΔT and strain ε can be obtained.
(7)$$\frac{\Delta \lambda }{\lambda }=(1-P_{e})\varepsilon +\left(\alpha +\xi \right)\Delta T\;$$
where $P_{e}$-photoelastic parameter of the optical fiber core for a general single-mode fiber, where $P_{e}\approx 0.22$.

α-thermal expansion of the optical fiber core.

ξ-thermal-optic coefficient of the optical fiber core.

Assume that the temperature change is zero, Equation (7) simplifies to
(8)$$\;\;\frac{\Delta \lambda }{\lambda }=(1-P_{e})\varepsilon =\left(1-P_{e}\right)\frac{\Delta x}{L}=\left(1-P_{e}\right)\frac{\frac{1}{2}H}{R+\frac{H}{2}}\;$$

In the case where the center wavelength of the FBG is known, H is the thickness of the silica gel and the FBG center wavelength drift Δλ can be measured from Equation (8). It follows that R is linearly related to $\frac{\Delta \lambda }{\lambda }$:
(9)$$R\propto a\cdot \frac{\lambda }{\Delta \lambda }\;$$
where a is a constant associated with the thickness, size and material of the shape sensor, the wavelength drifts of the fiber grating, etc., and the specific value is determined by testing.

### 2.2. Effect of Soft Substrate Material Size on Shape Sensors

There are few reports on the influence of the size of soft substrate shape sensors, i.e., the length, width and thickness of the soft material, on the performance. Few papers have analyzed the influence of FBG preburial depth; however, choosing the appropriate soft substrate material size is also one of the focuses of FBG shape sensor design. To analyze the influence of the length, width and thickness of the soft material on the performance of the FBG shape sensor, we choose common silica gel as the soft substrate and use the experimental model shown in Figure 2 to perform the following simulation and analysis in the workbench in the case where the size of the stainless steel, the direction and the position of the force are unchanged.

As shown in Figure 2, one end of the 304 stainless steel is fixed, the other free end is applied with a force F = 20 N, the direction is vertically downward and the silica gel is pasted on the stainless steel. The center always remains in the same line as the center of the stainless steel, regardless of the change in silica gel size. The dimensions of the stainless steel were as follows: Length of 100 mm, width of 25 mm, thickness of 10 mm. The length, width and thickness of the silica gel are indicated by L, D and H respectively.

#### 2.2.1. Variation of Silica Gel Length

When the width and thickness of the silica gel are constant, the length is changed and the stress of the silica gel is analyzed. The size changes of silica gel are shown in Table 1.

From Figure 3, we clearly find that the stress distribution of the silica gel varies significantly by length when the width and thickness of the silica gel are constant.

(1)If the length of the silica gel is longer, the stress area is larger and the maximum peak of the strain is greater, while the stress is concentrated in the upper layer of the neutral surface of the silica gel; if the length of the silica gel is smaller, the stress begins to shift from the upper layer to the lower layer of the neutral surface.(2)Regardless of the length of the silica gel, the stress distribution in the silica gel has left-right axis symmetry.

#### 2.2.2. Variation of Silica Gel Width

When the length and thickness of the silica gel are constant, the width of the silica gel is changed and the stress distribution of the silica gel is analyzed. The size changes of silica gel are shown in Table 2.

From Figure 4, we can find that the difference in silica gel stress distribution is small for different widths. When the width of the silica gel is varied, the maximum peak value of the strain of the silica gel varies from 0.0189 to 0.0186 MPa, with a small change in value, so we think that the effect of silica gel width on its stress can be ignored.

#### 2.2.3. Variation of Silica Gel Thickness

When the length and width of the silica gel are constant, the thickness of the silica gel is changed. Size changes of silica gel are shown in Table 3.

From Equation (9), we can deduce that if the thickness of the silica gel is larger, the measured radius of curvature is larger, but this is not actually the case, from Figure 5, we can find that the thickness changes and the silica gel stress distribution varies significantly when the silica gel length and width are constant.

(1)When the silica gel thickness changes, the maximum stress position of silica gel will also change, the thickness of the silica gel gradually increases, the area of the stress maximum gradually decreases, its position moves closer to the center of the silica gel, the gel thickness tends to the thickness of the stainless steel and the stress distribution is likely to be transformed from the upper layer to both the upper and lower layers.

Therefore, the thicker the silica gel thickness, the better the performance of the sensor; this method has not been established.

(2)The stress of silica gel reaches the maximum when the thickness is 5 mm, and the maximum stress increases with increasing thickness when the thickness is below 5 mm. The maximum stress decreases with increasing thickness when the thickness is above 5 mm.

In summary, selecting a reasonable silica gel size has a great role in improving the accuracy of the FBG shape sensor. The longer the length of the silica gel is, the larger the stress distribution area and the larger the maximum stress, while the width has less influence on the stress. Considering the practical application of shape sensors, the preparation process, the length of the grating of FBG and other factors, silica gel is selected as a rectangular structure with a length of 25–30 mm, width of 8–10 mm and thickness of 2–3 mm.

## 3. High-Precision FBG Shape Sensor Structure Design

Two main aspects should be considered in the design of high-precision FBG shape sensors based soft substrates: (1) The choice of the soft substrate—silica gel, as one of the most widely used and maturely processed soft materials, is suitable for the study of validation tests; (2) the layout problem of the grating in the silica gel. The influence of torque on the FBG soft substrate shape sensors cannot be ignored, and in order to minimize the influence of torque and improve the accuracy of the sensor, the FBG arrangement in Figure 6 is chosen.

The angle between the grating and the silica gel central axis is $\alpha_{1}$ and $\alpha_{2}$. We choose the simulation model shown in Figure 7 to simulate the stress of the FBG shape sensor when it is loaded with three loading methods, i.e., bending, torque and bending plus torque, and discuss the variation in the equivalent force of the FBG when the angles $\alpha_{1}$ and $\alpha_{2}$ are varied. All simulations are completed by Mechanical APDL software, and the equivalent stress stated in this section is the maximum equivalent stress on the nodes where the FBG is located.

In the simulation model shown in Figure 7, we set the silica gel size to a length L of 23 mm, width D of 8 mm and thickness H of 2 mm, while the length L_b_ and thickness H_b_ of the cantilever beam size are variable. The applied load is shown in Table 4.

### 3.1. The Stress Relevance between the Three Loading Methods

To ensure that the equivalent stress of the FBG is independent of the size of the cantilever beam and the location of the torque loaded, when three different loads are applied to the cantilever beam, we perform simulation and analysis for two different parameter settings, i.e., constant and variable beam size.

#### 3.1.1. Constant Beam Size (L_b_ = 500 mm H_b_ = 4 mm)

The loaded torque points are located at 1/4, 1/2 and 3/4 of the beam from the fixed end, and the equivalent stress simulation analysis of the FBG is shown in Table 5.

#### 3.1.2. Variable Beam Size

The length L_b_ and thickness H_b_ of the cantilever beam are variable, and the loaded torque points are located at 1/3 and 2/3 of the beam from the fixed end. The equivalent stress simulation of the FBG is shown in Table 6.

We analyze the tables of stress in Table 5 and Table 6 and obtain the following conclusions:
(1)When the cantilever beam is subjected to a separate torque, the maximum equivalent stress at the FBG on the silica gel is $F_{t}$; when the cantilever beam is subjected to a separate force, the maximum equivalent stress at the FBG on the silica gel is $F_{f}$; and when the cantilever beam is subjected to a combination of torque and force, the maximum equivalent stress at the FBG on the silica gel is $F_{m}$; Equation (10) is always established, which is independent of the position of the loaded torque and independent of the size of the cantilever beam.
(10)$$F_{t}+F_{f}=F_{m}$$(2)The value of the maximum equivalent stress at the FBG on the silica gel $F_{t}$ is affected by the size of the beam, the location of the loaded torque and other factors.


### 3.2. Influence of Angles $\alpha_{1}$ and $\alpha_{2}\;$ on Torque

In the experimental model shown in Figure 7, we set the cantilever beam length as 500 mm and thickness as 6 mm and load a force of 800 N vertically downward at the end of the beam and a torque of 800 N·m at 1/4 of the beam from the fixed end. In Mechanical APDL, we simulate and analyze the maximum equivalent stress at the nodes on four coordinate lines, which are the X-axis and Y-axis of the center of the silica gel, and the positions of the two FBGs are laid out. Simulation of equivalent stress on the 4-coordinate line at different values of α are shown in Figure 8.

From the simulation results, we can find the following characteristics:
(1)$\alpha_{1}=90{}^{\circ},\;\alpha_{2}=0{}^{\circ}:\;$The two FBGs correspond to the X-axis and Y-axis of the center of the silica gel, and the two maximum equivalent stress values at the positions of the two FBGs are 3.72 × 10^−13^ MPa and 1.92 × 10^−^^13^ MPa, respectively, which are generated by torque. These results show that the stress generated by torque is not sensitive to the X-axis and the Y-axis(2)$\alpha_{1}=\alpha_{2}\neq 0\neq 90{}^{\circ}$: The maximum equivalent stress at the nodes where FBGs is located are shown in Table 7.

In summary, the equivalent stress by torque has the following characteristics: When $\alpha_{1}$ is equal to 10° and 80°, 20° and 70°, 30° and 60°, or 40° and 50°, the maximum equivalent stress is equal for two different angles. With the increase of $\alpha_{1}$ and $\alpha_{2}$, the maximum equivalent stress increased and then decreases. The equivalent stress is the maximum when $\alpha_{1}$ and $\alpha_{2}$ are equal to 45°

The equivalent stress by force has the following characteristics: When $\alpha_{1}\;$is equal to 0°, the FBG is the most sensitive, and with the increase of $\alpha_{1}$ and $\alpha_{2}$, the maximum equivalent stress decreases and then increased.

For a comprehensive comparison, when $\alpha_{1}$ = $\alpha_{2}$ = 30°, as far as the equivalent stresses by the three loads at the FBGs are concerned, the stress by force is large and the stress by torque is relatively pronounced. Therefore, the parameters of the dual FBG high-precision shape sensor are set as follows: 

The length is 23 mm, the width is 8 mm, the thickness is 2 mm, angle $\alpha_{1}=\alpha_{2}$= 30°, the FBG length is 5 mm and the depth that the FBGs are preburied is 0.5 mm from the neutral plane in the silica gel.

## 4. Sensor Performance Testing

### 4.1. Fabrication of the Dual FBG High-Precision Shape Sensor 

The silica gel in this paper is a translucent silicone, liquid, and it must be mixed with a certain proportion of curing agent evenly to become a solid state as shown in Figure 9. The cured silica gel product is high temperature resistant, tough and tear resistant. After many tests, it is discovered that the optimal ratio of silica gel and curing agent is 15 g:0.1 g– ~ 10 g:0.1 g, which can not only avoid the silica gel being cured in the process of vacuuming, but also ensure that the curing time at normal temperature does not exceed 8 h, which is shorter in the 90 °C heating environment.

A large number of bubbles will be generated during the mixing process of silica gel and curing agent. If there are a large number of bubbles in the curved shape sensor, it will affect the stress transfer between FBG and silica gel, and reduce the sensitivity of the sensor, so it is necessary to vacuum before the silica gel is injected into the mold, as shown in Figure 10. 

The sensor curing process and the finished sensor after curing are shown in Figure 11.

### 4.2. FBG Shape Sensor Test Platform 

The shape sensor designed in this paper is mainly used for shape reconstruction of objects similar to wing structures, so the cantilever beam is used for parameter calibration, especially when the influence of torque on the accuracy of the shape sensor is considered. The traditional cylinder calibration method is not used because in the reconstruction process, we mainly consider the relationship between the rotation angle of the reconstructed object and the FBG wavelength drift, rather than the relationship between the size of the object subjected to torque and wavelength drift. The performance parameters of the dual FBG high-precision shape sensor were tested by building the sensor test platform shown in Figure 12.

The material of the beam is stainless steel, the length of the beam is 500 mm and three types of sensors are attached at 200 mm from the fixed end of the beam: Single FBG silica gel shape sensor (labeled FBG2) on the left, bare FBG (labeled FBG1) in the middle and dual FBG silica gel shape sensor (labeled FBG3/FBG4) on the right. The parameters of all FBGs are shown in Table 8

The test experiment was divided into several groups, and each group of experimental processes was repeated many times. By processing the data recorded in the experiment, various performance parameters of this sensor, such as sensitivity, repeatability error and hysteresis error, can be analyzed.

### 4.3. Sensor Parameter Calibration

#### 4.3.1. Curvature Calibration Loaded with Vertical Force

At the free end of the beam, weights of 101 g, 175 g, 175 g, 175 g, 500 g, 500 g and 500 g were added in turn, and the reading of the corresponding FBG high-speed demodulator wavelength for each weight was recorded one by one.

The relationship between the curvature k of any point on the curve and the derivative of the curve equation w is shown in Equation (11).
(11)$$k=\pm \frac{w^{\prime \prime }}{{\left(1+{w^{\prime }}^{2}\right)}^{\frac{3}{2}}}$$

For cantilever beams, the curve equation w is the deflection curve of the beam after the beam has been subjected to the force, which has the following physical relationship with the curvature k and the bending moment M(x):(12)$$\frac{w^{\prime \prime }}{{\left(1+{w^{\prime }}^{2}\right)}^{\frac{3}{2}}}=-\frac{M\left(x\right)}{EI}\;$$

In this experiment, the deflection curve of the beam is a flat curve, and ${w^{\prime }}^{2}\ll 1$, which is negligible, Equation (12) is simplified as
(13)$$w^{\prime \prime }=-\frac{M\left(x\right)}{EI}$$

Equation (14) can be obtained from the Equations (11) and (12).
(14)$$k=-\frac{M}{EI}$$

The bending moment equation M(x) of any cross section at a distance *x* from the fixed end of the cantilever beam can be expressed as Equation (15).
(15)M(x)=−F(l−x) 

Here: l-length of the cantilever beam, where l=0.4 m

x-distance between sensor center and fixed end, where x=0.2 m

EI-cantilever beam flexural stiffness, where $EI=5.0333\;N\cdot m^{2}$

The curvature k of the cantilever beam at a distance *x* from the fixed end can be derived from Equations (14) and (15)
(16)$$k=-\frac{M}{EI}=\frac{F(l-x)}{EI}\;$$

The sensitivity of FBG sensors refers to the ratio of the change in the input physical quantity to the change in the output center wavelength when the sensor is in a steady state. Therefore, the sensitivity of three types of FBG sensors can be expressed as Equation (17).
(17)$$S=\frac{\Delta \lambda }{\Delta k}$$

It can be judged from Figure 13 and Table 9 that for the dual FBG high-precision shape sensor loaded with vertical force, (1) the center wavelength drift of FBG2 and FBG3 is linearly related to the curvature change; (2) the sensitivity of FBG3 and FBG3 is 54~55 pm per 1 m^−1^, which is lower than that of bare FBG1 and the single FBG shape sensor FBG2.

#### 4.3.2. Calibration of the Rotation Radian Loaded with Torque

The cantilever beam will flip under the action of torque. In the test model shown in Figure 12, first, the fixture shown in the figure is welded on the cantilever beam, then torque is applied to the cantilever beam using a torque wrench, and the displacement of the edge of the beam to the position of the shape sensor is measured. The cantilever beam is in equilibrium when no torque is applied, and the cantilever beam is gradually loaded with torque to ensure that the micrometer degree increases by 0.5 mm in turn, i.e., the beam rotates by 0.021 radians with one torque loaded.

The wavelength drift of the bare grid, where the grid is at the center of the cantilever beam axis, is almost 0. The relationships between the radian and wavelength drift for the single grid shape sensor (FBG2) and the dual FBG shape sensors (FBG3 and FBG4) are shown in Figure 14.

It can be judged from Figure 14 and Table 10 that for the dual FBG high-precision shape sensor loaded torque, (1) the center wavelength drift of FBG3 and FBG4 is linearly related to the radian change; (2) the sensitivity of FBG3 and FBG4 is 866.5~899 pm per 1, which is more than three times the sensitivity of the single FBG shape sensor FBG2.

#### 4.3.3. Calibration of the Dual FBG High-Precision Shape Sensor under a Combined Load

From the above analysis, it can be discovered that the sensitivity of FBG3 and FBG4 in the dual FBG high-precision shape sensor differs when different loads are applied.

When a vertical force is applied, the sensitivity to curvature is as follows
$$S_{k3}=53.57\mathrm{pm}/m^{-1}\;S_{k4}=54.69\mathrm{pm}/m^{-1}$$

The corresponding wavelength drift is shown in Equations (18) and (19), where Δk is the curvature change.
(18)$$\Delta \lambda_{31}=S_{k3}*\Delta k$$
(19)$$\Delta \lambda_{41}=S_{k4}*\Delta k\;$$

When a torque is applied, the sensitivity to the rotation radian is as follows:$$S_{r3}=898.956\mathrm{pm}/1\;S_{r4}=-866.46\mathrm{pm}/1$$

The corresponding wavelength drift is shown in Equations (20) and (21), where Δr is the rotation radian change.
(20)$$\;\Delta \lambda_{32}=S_{r3}*\Delta r$$
(21)$$\Delta \lambda_{42}=S_{r4}*\Delta r\;$$

The forces of the two different loads mentioned above can be superimposed in the dual FBG high-precision shape sensor.

From Equations (18)–(21), we can obtain
(22)$$\Delta \lambda_{3}=S_{k3}*\Delta k+S_{r3}*\Delta r\;$$
(23)$$\Delta \lambda_{4}=S_{k4}*\Delta k+S_{r4}*\Delta r\;\;$$

The curvature change Δk and the rotation radian change Δr in the dual FBG high-precision shape sensor are expressed as Equations (24) and (25), when the combined loads are applied.
(24)$$\Delta k=\frac{S_{r4}*\Delta \lambda_{3}-S_{r3*}\Delta \lambda_{4}}{S_{k3}*S_{r4}-S_{k4}*S_{r3}}\;$$
(25)$$\Delta r=\frac{S_{k4}*\Delta \lambda_{3}-S_{k3*}\Delta \lambda_{4}}{S_{k4}*S_{r3}-S_{k3}*S_{r4}}$$

### 4.4. Error Analysis of Repeatability

On the experimental platform shown as Figure 12, a torque of 8 N·m is loaded on the cantilever beam at the free end of the beam, and the mass block is added in turn with weights of 101 g, 175 g, 175 g, 175 g, 175 g, and 500 g. We repeat five tests and record the relationship between the center wavelength drift of FBG3 and FBG4 and the change in curvature to obtain the curves shown in Figure 15.

The repeatability errors *R* of FBG3 and FBG4 are calculated from the sensor error formula of repeatability:$$R3=\frac{\Delta \lambda }{{\left(\Delta \lambda \right)}_{FS}}\times 100\%=\frac{2}{123}\times 100\%=1.63\%\;R4=\frac{\Delta \lambda }{{\left(\Delta \lambda \right)}_{FS}}\times 100\%=\frac{2}{123}\times 100\%=1.63\%$$

Here: $\Delta \lambda_{max}$-maximum deviation value of the FBG center wavelength drift in forward-reverse stroke.

${\left(\Delta \lambda \right)}_{FS}$ -the center wavelength drift of the FBG output at full scale

For the above repeatability test results, using the same operation procedure, we randomly selected a shape sensor for the test as shown in Figure 16, two of FBGs in the sensor are labeled as FBG1 and FBG2, respectively.

If the sensor has multiple output characteristic curves under the same input conditions many times, the sensor has better repeatability and smaller error, which indicates that the FBG center wavelength drift in the dual FBG high-precision shape sensor is more stable.

### 4.5. Hysteresis Error Analysis

Hysteresis error is one of the important indicators to reflect the accuracy of the sensor. The hysteresis of the sensor is easily caused by the physical properties of the sensor’s sensitive component materials or design defects.

On the experimental platform shown as Figure 12, a torque of 8 N·m is loaded on the cantilever beam at the free end of the beam. First, the mass block is increased in turn with weights of 101 g, 175 g, 175 g, 175 g, 175 g and 500 g. Then, the mass block is sequentially decreased, with weights of 500 g, 175 g, 175 g, 175 g, 175 g and 101 g. We repeat five tests and record the relationship between the center wavelength drift of FBG3 and FBG4 and the change of curvature. The obtained curves are shown in Figure 17.

The hysteresis errors of FBG3 and FBG4 are calculated from the hysteresis error formula.
$$\;e_{H3}=\frac{\Delta \lambda_{max}}{{\left(\Delta \lambda \right)}_{FS}}\times 100\%=\frac{3}{123}\times 100\%=2.43\%\;e_{H4}=\frac{\Delta \lambda_{max}}{{\left(\Delta \lambda \right)}_{FS}}\times 100\%=\frac{3}{123}\times 100\%=2.43\%$$

Here: $\Delta \lambda_{max}$-maximum deviation value of FBG center wavelength drift in forward-reverse stroke.

${\left(\Delta \lambda \right)}_{FS}$-the center wavelength drift of the FBG output at full scale.

For the above hysteresis test results, using the same operation procedure, we randomly selected a shape sensor for the test as shown in Figure 18, two of FBGs in the sensor are labeled as FBG1 and FBG2, respectively.

From the hysteresis error in Figure 16, we can intuitively judge that the FBG center wavelength drift of the dual FBG high-precision shape sensor is stable, and the operating performance of the dual FBG high-precision shape sensor is also stable.

## 5. Shape Reconstruction Test of a Test Specimen

To verify the high accuracy and reliability of the dual FBG high-precision shape sensor in the process of shape reconstruction, which is proposed in this paper, we will carry out a reconstruction test on a test specimen installed with 19 shape sensors (38 FBGs in total), and the experimental setup is shown in Figure 19.

According to the FBG center wavelength drift in each dual FBG high-precision shape sensor, the discrete curvature information of each measurement point is obtained, and the improved Kalman filtering algorithm is used. The three-dimensional shape of the test specimen is reconstructed by the curvature, coordinate position and other information of the measured point in MATLAB software. The loading method of the test specimen is shown in Figure 18.

In Figure 20a, a weight mass is added in the middle loading point, and it is added three times in turn. In Figure 20b, a weight mass is added at the left loading point, and an upward force is loaded at the right loading point. The reconstructing shapes of the specimen are shown in Figure 20 and Figure 21.

In Figure 21 and Figure 22, the change value of the measurement point in the Z-axis coordinate reflects the degree of specimen bending, which corresponds to the change in curvature of the dual FBG shape sensor, and the change in the x-axis coordinate reflects the degree of specimen rotation, which corresponds to the change in the rotation angle in the dual FBG shape sensor. To verify the accuracy of the shape reconstruction, comparing the measured and reconstructed point coordinates of the FBG shape sensor with the actual coordinates of two measurement points is shown as Table 11 and Table 12. (This results comparison ignores the error caused by the reconstruction algorithm).

As shown in Table 11 and Table 12, the maximum reconstruction error is 6.13% (z-axis) when the test specimen is reconstructed in shape, and the greater the curvature, the smaller the reconstruction error. The results show that the shape reconstruction of the test specimen based on the dual FBG shape sensor is highly restored, and the dual FBG shape sensor has the high feasibility and effectiveness in 3D shape reconstruction.

## 6. Conclusions

To improve the measurement accuracy of FBG shape sensors, especially in terms of structural shape reconstruction accuracy, the design idea of a high-precision shape sensor based on dual FBGs is proposed. First, a high-precision shape sensor with dual FBGs is designed and analyzed. Second, the linear relationship among the FBG wavelength drift and curvature, rotation radian loaded vertical force and torque is obtained. Finally, the verification test of shape reconstruction is completed in the laboratory. The test specimen is loaded with an arbitrary load, and the discrete curvature is collected according to the dual FBG high-precision shape sensor. Then the three-dimensional shape of the test specimen is reconstructed by using the reconstruction algorithm. Experimental results show that the maximum reconstruction error of the sensor proposed in the paper is less than 6.5%.

In summary, we can draw the following conclusion: A dual FBG high-precision shape sensor is an effective method for improving structural shape reconstruction, especially in the field of deformation monitoring such as robots, smart wearables, wings and fan blades, and has greater application potential. In the future, we will increase the sensitivity of the dual FBG high-precision shape sensor, improve the shape reconstruction algorithm and further improve the three-dimensional shape reconstruction precision of the structural deformation.

## Figures and Tables

**Figure 1 sensors-22-00168-f001:**
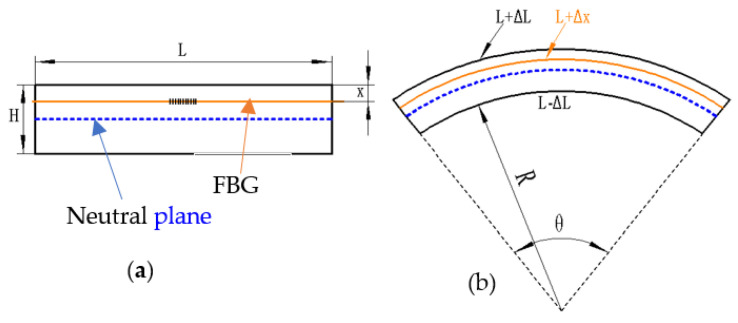
Bending mechanism of the elastic material: (**a**) Stress-free without bending, (**b**) stress and bend.

**Figure 2 sensors-22-00168-f002:**
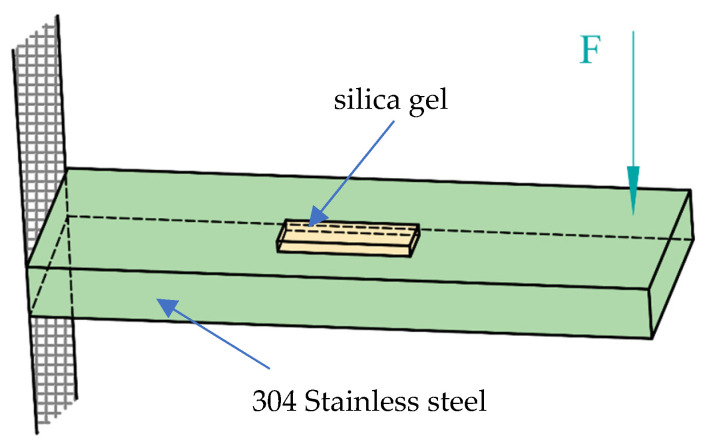
Sensor simulation model.

**Figure 3 sensors-22-00168-f003:**
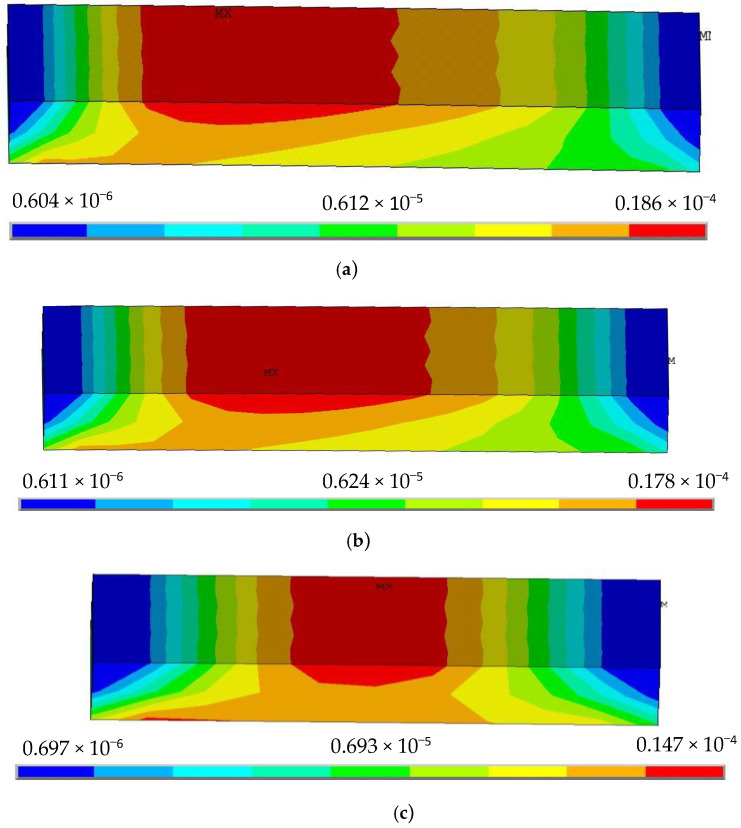
Stress distribution nephogram of silica gel with different lengths, (**a**) L = 30 mm, (**b**) L = 25 mm, (**c**) L = 15 mm, (**d**) L = 8 mm.

**Figure 4 sensors-22-00168-f004:**
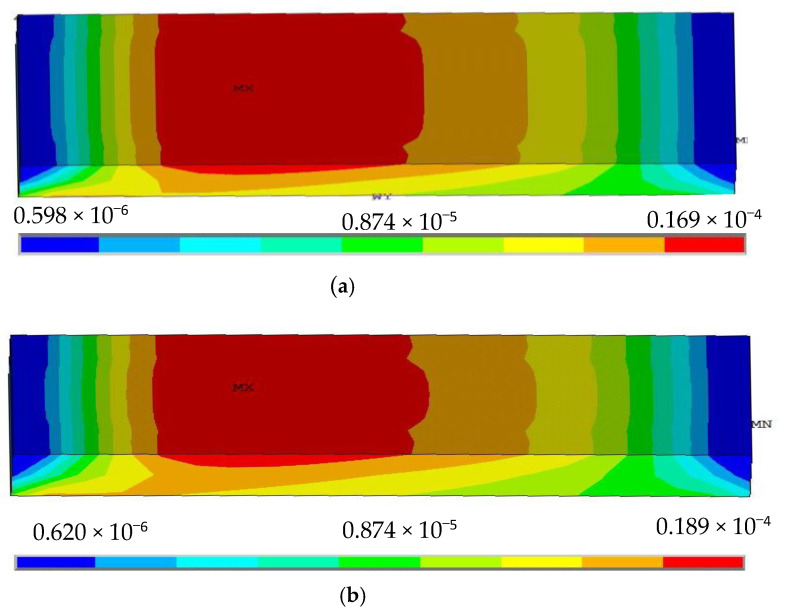
Stress distribution nephogram of silica gel with different widths: (**a**) D = 25 mm, (**b**) D = 15 mm, (**c**) D = 10 mm, (**d**) D = 8 mm.

**Figure 5 sensors-22-00168-f005:**
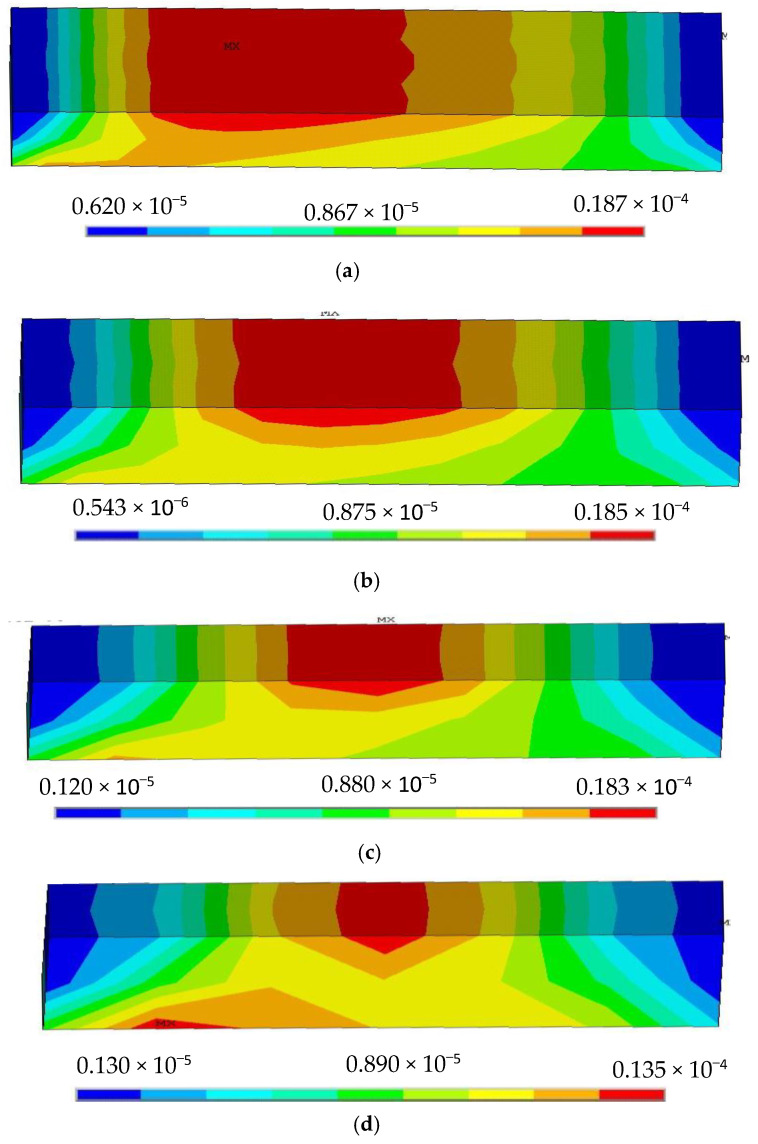
Stress distribution nephogram of silica gel with different thickness: (**a**) H = 2 mm, (**b**) H = 5 mm, (**c**) H = 8 mm, (**d**) H = 10 mm.

**Figure 6 sensors-22-00168-f006:**
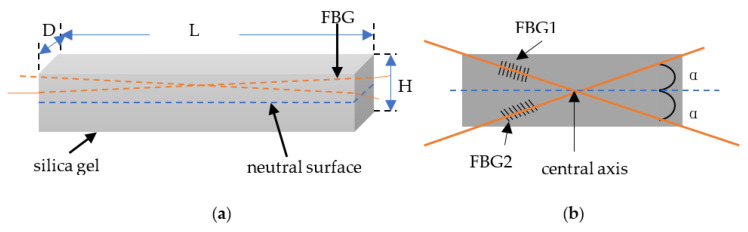
Structure diagram of the dual FBG high-precision shape sensor: (**a**) Dual FBG shape sensor structure Figure, (**b**) FBG layout Figure.

**Figure 7 sensors-22-00168-f007:**
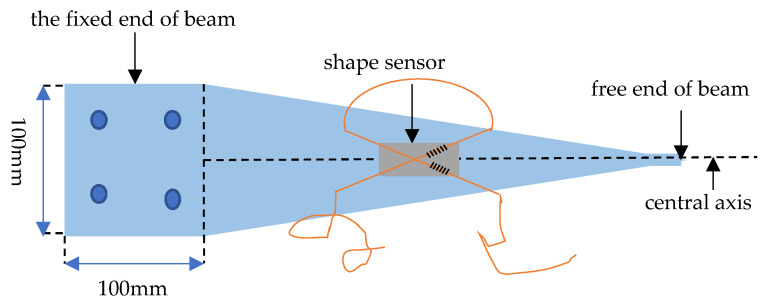
Simulation model of dual FBG high-precision shape sensor.

**Figure 8 sensors-22-00168-f008:**
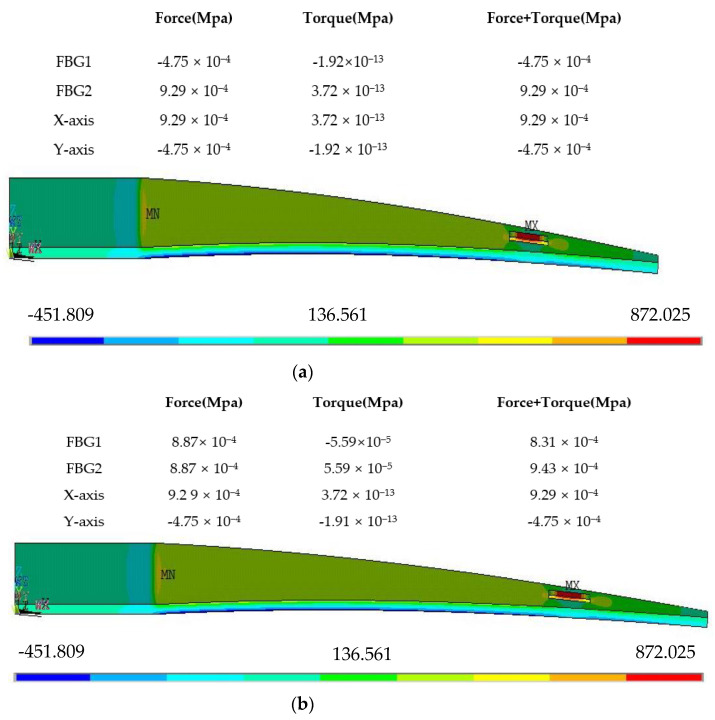
Simulation of equivalent stress on the 4-coordinate line at different values of α, (**a**) $\alpha_{1}=90{}^{\circ}\alpha_{2}=0{}^{\circ}$, (**b**) $\;\alpha_{1}=\alpha_{2}=10$°.

**Figure 9 sensors-22-00168-f009:**
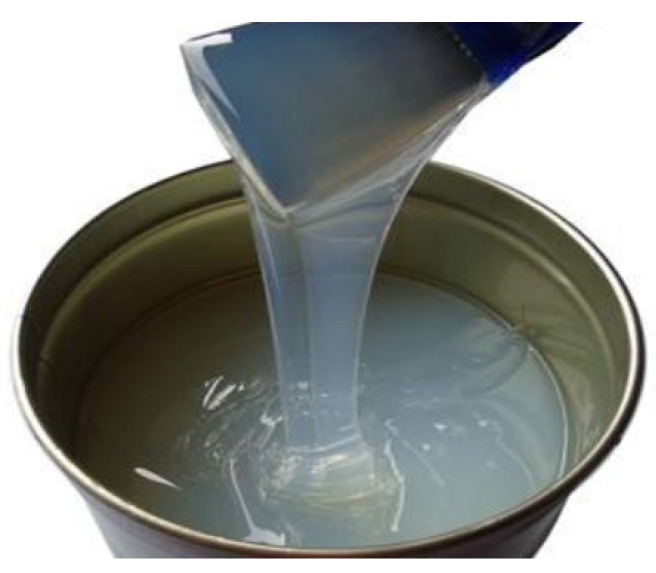
Liquid-like translucent silica gel.

**Figure 10 sensors-22-00168-f010:**
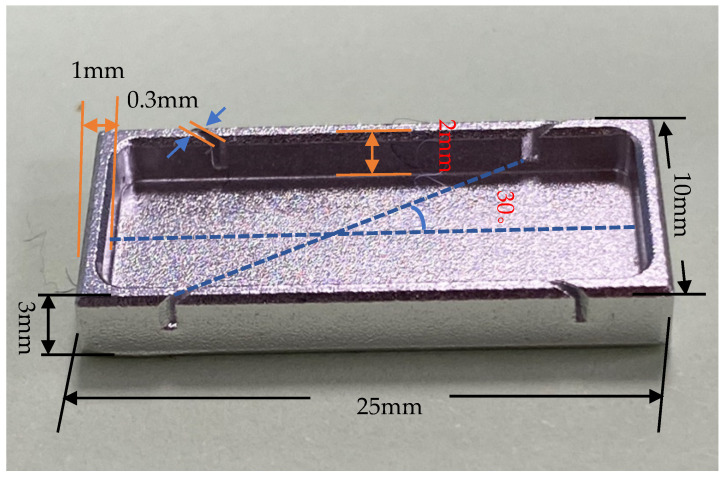
Mold of Sensor Fabrication.

**Figure 11 sensors-22-00168-f011:**
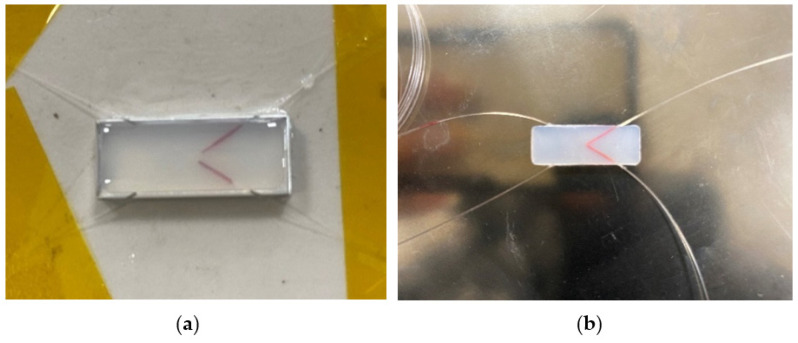
Fabrication of Dual FBG High-Precision Shape sensor, (**a**) Sensor curing process, (**b**) Sensor sample.

**Figure 12 sensors-22-00168-f012:**
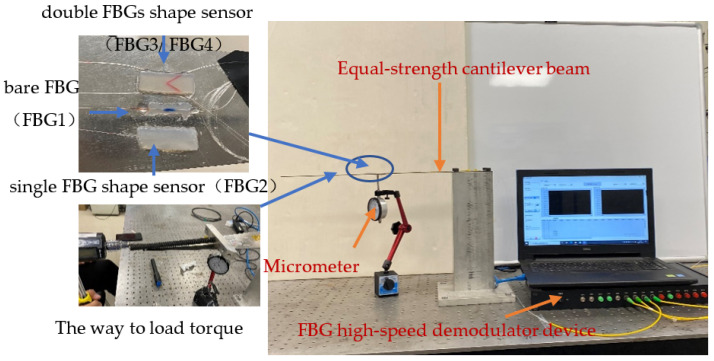
Dual FBG High-Precision Shape Sensor Test Platform.

**Figure 13 sensors-22-00168-f013:**
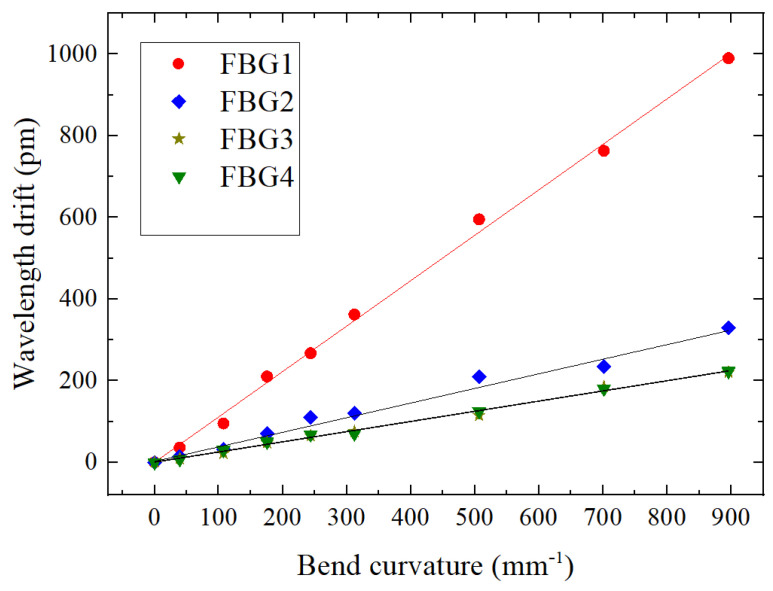
Relationship between bending and wavelength drift of three kinds of FBG sensors.

**Figure 14 sensors-22-00168-f014:**
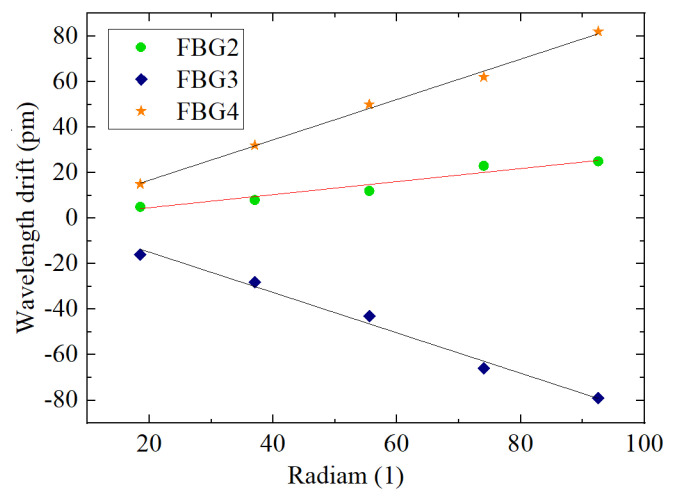
Relationship between radian and FBG wavelength drift of FBG shape sensor.

**Figure 15 sensors-22-00168-f015:**
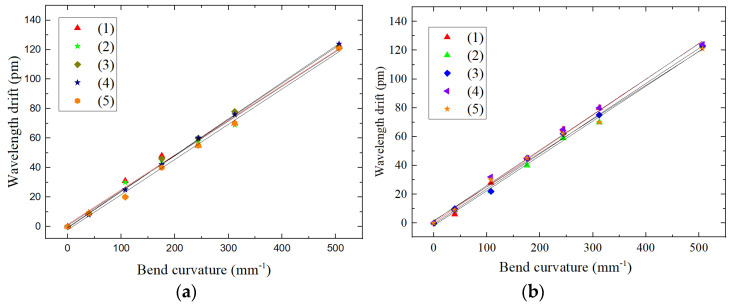
Shape sensor FBG error analysis of repeatability: (**a**) FBG3 center wavelength drift consistency, (**b**) FBG4 center wavelength drift consistency.

**Figure 16 sensors-22-00168-f016:**
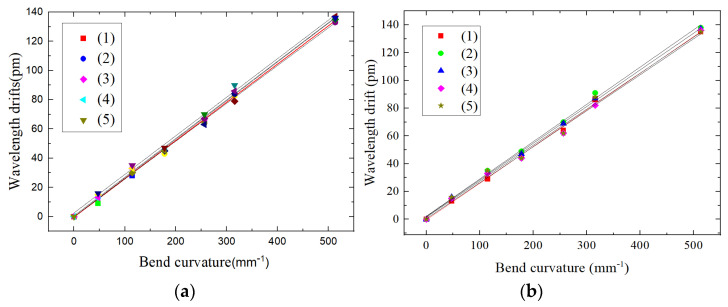
Shape sensor FBG error analysis of repeatability: (**a**) FBG1 center wavelength drift consistency, (**b**) FBG2 center wavelength drift consistency.

**Figure 17 sensors-22-00168-f017:**
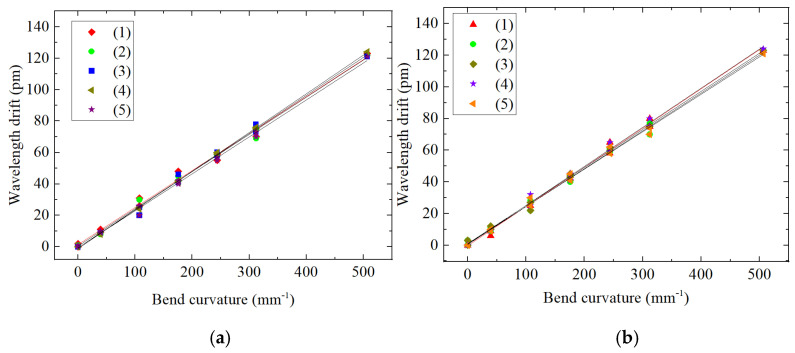
Shape sensor FBG hysteresis error analysis: (**a**) FBG3 center wavelength drift in forward-reverse stroke, (**b**) FBG4 center wavelength drift in forward-reverse stroke.

**Figure 18 sensors-22-00168-f018:**
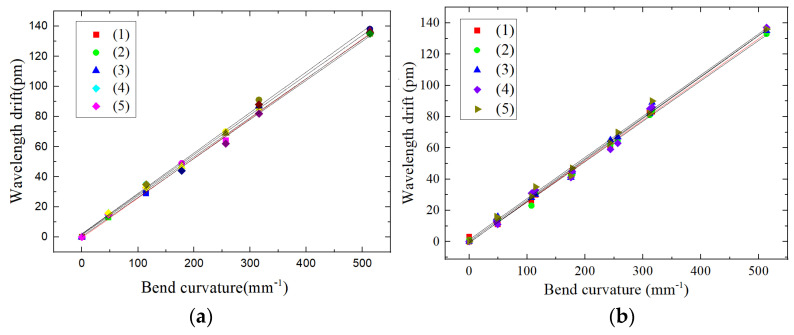
Shape sensor FBG hysteresis error analysis: (**a**) FBG1 center wavelength drift in forward-reverse stroke, (**b**) FBG2 center wavelength drift in forward-reverse stroke.

**Figure 19 sensors-22-00168-f019:**
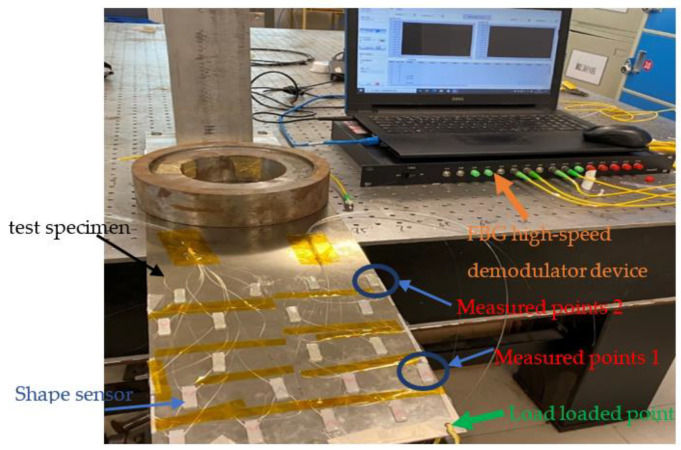
Shape sensor reconstruction experiment platform.

**Figure 20 sensors-22-00168-f020:**
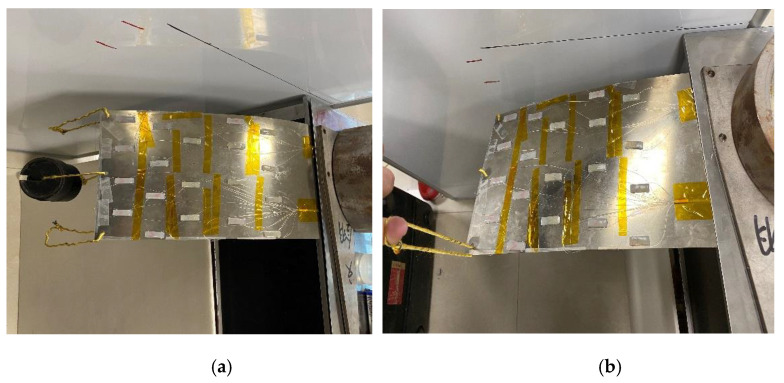
Method of test piece loaded, (**a**) Load in the middle point, (**b**) Load in both side points.

**Figure 21 sensors-22-00168-f021:**
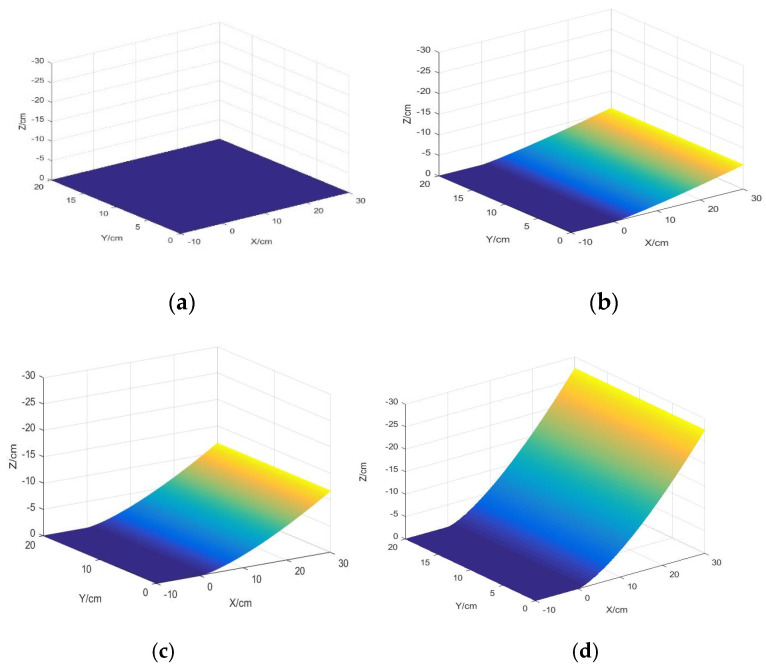
Test piece shape reconstruction loaded in the middle point: (**a**) Initial state, (**b**) state I, (**c**) state II, (**d**) state III.

**Figure 22 sensors-22-00168-f022:**
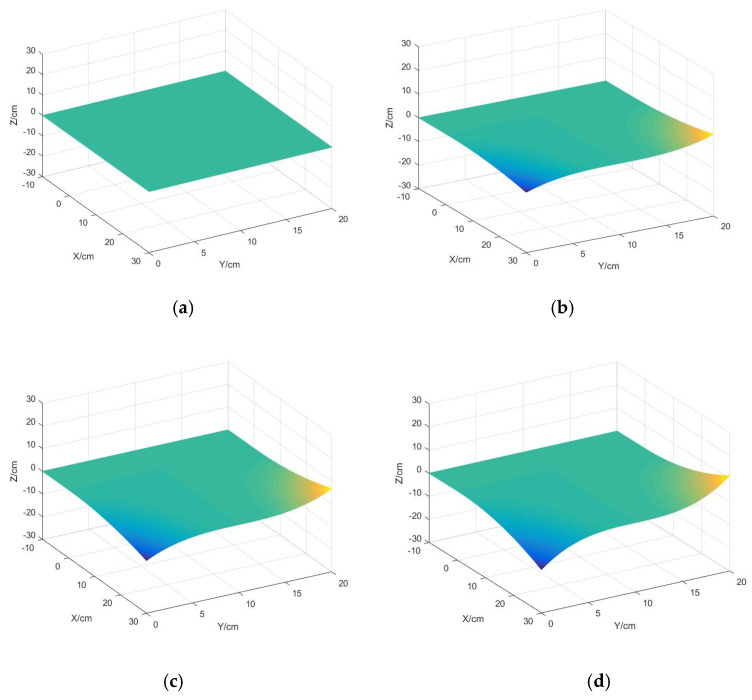
Test piece shape reconstruction loaded in both side points: (**a**) Initial state, (**b**) state IV, (**c**) state V, (**d**) state VI.

**Table 1 sensors-22-00168-t001:** Variation of silica gel length.

Size	a	b	c	d
length (mm)	30	25	15	8
width (mm)	8	8	8	8
thickness (mm)	3	3	3	3

**Table 2 sensors-22-00168-t002:** Variation of the silica gel width.

Size	a	b	c	d
Length (mm)	30	30	30	30
Width (mm)	25	15	10	8
Thickness (mm)	3	3	3	3

**Table 3 sensors-22-00168-t003:** Variation of the silica gel thickness.

Size	a	b	c	d
length (mm)	30	30	30	30
width (mm)	10	10	10	10
thickness (mm)	2	5	8	10

**Table 4 sensors-22-00168-t004:** Parameters of the applied load.

Parameter	Force	Torques	Force + Torques	
value	800 N	800 N·m	800 N	800 N·m
direction	vertical down	clockwise	vertical down	clockwise
loading point	End of beam (free end)	Any position of beam	End of beam (free end)	Any position of beam

**Table 5 sensors-22-00168-t005:** Absolute value of the maximum equivalent stress.

Loaded	Force		Torque		Force + Torque	
Point	FBG1 (MPa)	FBG2 (MPa)	FBG1 (MPa)	FBG1 (MPa)	FBG2 (MPa)	FBG1 (MPa)
1/4	5.78 × 10^−4^	5.78 × 10^−4^	−1.80 × 10^−11^	1.82 × 10^−11^	5.78 × 10^−4^	5.78 × 10^−4^
1/2	5.78 × 10^−4^	5.78 × 10^−4^	−1.04 × 10^−8^	1.04 × 10^−8^	5.78 × 10^−4^	5.78 × 10^−4^
3/4	5.78 × 10^−4^	5.78 × 10^−4^	−1.41 × 10^−4^	1.41 × 10^−4^	4.37 × 10^−4^	7.19 × 10^−4^

**Table 6 sensors-22-00168-t006:** Absolute value of the maximum equivalent stress.

Size/	Force		Torque		Force +Torque	
Loaded Point	FBG1(MPa)	FBG2 (MPa)	FBG1 (MPa)	FBG2 (MPa)	FBG1 (MPa)	FBG2 (MPa)
L_b_ = 300 mm, H_b_ = 4 mm,1/3	1.12 × 10^−4^	1.12 × 10^−4^	−5.45 × 10^−8^	5.45 × 10^−8^	1.12 × 10^−4^	1.12 × 10^−4^
L_b_ = 400 mm, H_b_ = 4 mm,1/3	1.86 × 10^−4^	1.86 × 10^−4^	−3.62 × 10^−10^	3.62 × 10^−10^	1.86 × 10^−4^	1.86 × 10^−4^
L_b_ = 500 mm, H_b_ = 5 mm,2/3	1.13 × 10^−4^	1.13 × 10^−4^	−5.45 × 10^−8^	5.45 × 10^−8^	1.13 × 10^−4^	1.13 × 10^−4^
L_b_ = 600 mm, H_b_ = 8 mm,2/3	2.49 × 10^−4^	2.49 × 10^−4^	−3.57 × 10^−7^	3.57 × 10^−7^	2.49 × 10^−4^	2.49 × 10^−4^

**Table 7 sensors-22-00168-t007:** Absolute value of the maximum equivalent stress.

$\alpha_{1}$$/\alpha_{2}$ (°)	Stress_(torque)_ (MPa)	Stress_(force)_ (MPa)	$\alpha_{1}$$/\alpha_{2}$ (°)	Stress_(torque)_ (MPa)	Stress_(force)_ (MPa)
$\alpha_{1}$= 0 $\alpha_{2}$ = 90	----	----	$\alpha_{1}=\alpha_{2}$ = 45	1.63 × 10^−4^	2.27 × 10^−4^
$\alpha_{1}=\alpha_{2}$ = 10	5.59 × 10^−5^	8.87 × 10^−4^	$\alpha_{1}=\alpha_{2}$ = 50	1.61 × 10^−4^	1.05 × 10^−4^
$\alpha_{1}=\alpha_{2}$ = 20	1.05 × 10^−4^	7.65 × 10^−4^	$\alpha_{1}=\alpha_{2}$ = 60	1.41 × 10^−4^	1.24 × 10^−4^
$\alpha_{1}=\alpha_{2}$ = 30	1.41 × 10^−4^	5.78 × 10^−4^	$\alpha_{1}=\alpha_{2}$ = 70	1.05 × 10^−5^	3.14 × 10^−4^
$\alpha_{1}=\alpha_{2}$ = 40	1.61 × 10^−4^	3.49 × 10^−4^	$\alpha_{1}=\alpha_{2}$ = 80	5.59 × 10^−5^	4.32 × 10^−4^

**Table 8 sensors-22-00168-t008:** The detailed parameters of the FBG sensors.

Parameter	FBG1	FBG2	FBG4	FBG3
	(Bare FBG)	(Single FBG Sensor)	(Dual FBG Shape Sensor)	
Fiber type	single mode 1549.190 5	single mode 1536.141 5	single mode 1529.953 5	single mode 1536.185 5
Wavelength (nm)				
Grating length (mm)				
Peak reflectivity (%)	75	75	75	75

**Table 9 sensors-22-00168-t009:** Sensitivity of the three kinds of FBG sensors.

Parameters	FBG1	FBG2	FBG4	FBG3
Δλ (pm)	217	70	49	48
Δk (m^−1^)	0.896024	0.896024	0.896024	0.896024
*S_k_* (pm/m^−1^)	242.18	78.12	54.69	53.57

**Table 10 sensors-22-00168-t010:** Radian sensitivity of the two kinds of FBG sensors.

Parameters	FBG2	FBG4	FBG3
Δλ (pm)	25	−80	83
Δr (1)	0.09233	0.09233	0.09233
*Sr* (pm/1)	270.77	−866.46	898.956

**Table 11 sensors-22-00168-t011:** Comparison of the results of the measurement points in Figure 21.

Measured Points	Parameters	State I	State II	State III
point 1	Reconstructed value(cm)	1.582	2.497	5.868
	Measured value(cm)	1.671	2.57	6.13
	Error (%)	5.33	2.84	4.27
point 2	Reconstructed value(cm)	4.65	8.566	19.52
	Measured value(cm)	4.51	8.151	20.58
	Error (%)	3.1	5.09	5.15

**Table 12 sensors-22-00168-t012:** Comparison of the results of the measurement points in Figure 22.

Measured Points	Parameters	State IV		State V		State VI	
		x-axi	z-axis	x-axi	z-axis	x-axi	z-axis
point 1	Reconstructed values (cm)	9.917	0.497	9.899	0.6798	9.891	1.254
	Measured values (cm)	9.751	0.468	9.741	0.722	9.732	1.198
	Error (%)	1.7	6.13	1.62	5.84	1.63	4.67
point 2	Reconstructed values (cm)	24.51	2.848	23.99	4.48	23.08	7.074
	Measured values (cm)	24.109	2.769	23.465	4.356	22.591	7.267
	Error (%)	1.66	2.85	2.24	2.85	2.16	2.66

## Data Availability

Not applicable.

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
