# Peer review of "Design and Measurement of a Dual FBG High-Precision Shape Sensor for Wing Shape Reconstruction"

_sensors, 2021, doi:10.3390/s22010168_

Round 1
Reviewer 1 Report
FBG shape sensors can be applied in the wing shape sensing and reconstruction for the aircrafts. Wu et al presented a soft substrates shape sensor using the dual FBGs. By numerical modeling, the parameter optimization of the soft substrate was figured out, e.g. the length, width, and thickness. The dual FBG cross-laying method is adopted. The best crossover angle between the FBGs 60Ois selected as the most sensitive angle to the torque response. The calibration test showed that the sensor exhibited the linear relationship between the FBG wavelength shift and curvature, rotation radian loaded vertical force and torque. Real experimental results validated the analysis of the sensor design. The work is well designed and clearly described. It can be published after addressing several minor issues.
Comments:
- Abstract, “where soft substrates and torque are two important factors”. The “soft substrate” here is too general to understand the motivation of the work. Please specify it.
- Abstract, “effectively reduce the influence of torque” and “α = 30°is selected as the most sensitive angle to the torquer response” the two sentences contradict each other. Please double check the description.
- Introduction, please uniform the author names of the literature, e.g. “Wang” and “Guanhua Zhi”
- Line 221, the ranges of the optimized parameters seem a little larger for a simulation result.
- Table 8, please add the description on the roles of each FBG.
Several English errors should be avoided, to name a few
- Line 88, “on the one hand”
- Line 120, “neutral surface”
Author Response
Dear Editor,
Thank you very much for your kind works to our manuscript titled
Design and Measurement of a Dual FBG High-Precision Shape Sensor for Wing Shape Reconstruction
This manuscript was revised carefully according to the comments by the reviewers, so we hope that this revised version is satisfactory.
Best regards!
Sincerely yours,
Huifeng Wu
-----------------------------------------------------------------------
Reviewer: 1
Comments:
FBG shape sensors can be applied in the wing shape sensing and reconstruction for the aircrafts. Wu et al presented a soft substrates shape sensor using the dual FBGs. By numerical modeling, the parameter optimization of the soft substrate was figured out, e.g. the length, width, and thickness. The dual FBG cross-laying method is adopted. The best crossover angle between the FBGs 60 is selected as the most sensitive angle to the torque response. The calibration test showed that the sensor exhibited the linear relationship between the FBG wavelength shift and curvature, rotation radian loaded vertical force and torque. Real experimental results validated the analysis of the sensor design. The work is well designed and clearly described. It can be published after addressing several minor issues.
- Abstract, “where soft substrates and torque are two important factors”. The “soft substrate” here is too general to understand the motivation of the work. Please specify it.
Response:
The soft substrate has a significant influence on the performance of the sensor, including, substrate material differences, size. Different soft substrate materials are described in the literature, and the purpose of shape sensor is different, such as deformation monitoring of airplane skin, wings, etc. [4-6], smart wearable devices [10-11]. In particular, the effect of the size of the soft substrate on sensor performance. In Section 2.2, we introduced the effect of the dimensions of the soft substrate, including influence of the length, width and thickness of the soft material on the performance of the FBG shape sensor. Figure 3 shows the stress distribution of the silica gel varies significantly by length when the width and thickness of the silica gel are constant. If the length of the silica gel is longer, the stress area is larger and the maximum peak of the strain is greater. Figure 4 shows the difference in silica gel stress distribution is small for different widths. Figure 5 shows the thickness changes and the silica gel stress distribution varies significantly. When the silica gel thickness changes, the maximum stress position of silica gel will also change, the thickness of the silica gel gradually increases, the area of the stress maximum gradually decreases. Therefore, we think that “soft substrates and torque are two important factors” needs be highlighted.
- Abstract, “effectively reduce the influence of torque” and “α = 30°is selected as the most sensitive angle to the torquer response” the two sentences contradict each other. Please double check the description
Response:
Page 1: “effectively reduce the influence of torque” ----- The meaning we want to express is:
The Sensors with two FBG cross-laying method can measure the wavelength drift caused by bending and torque at the same time and count separately, traditional shape sensors superimpose wavelength drift by torque in wavelength drift by bending.
“α = 30°is selected as the most sensitive angle to the torquer response” ----- The meaning we want to express is:
Under the same loading conditions, the sensors with this crossover angle can measure the maximum wavelength drift by torque. In Section 2.2, we gave a more detailed description of the effect of crossover angles.
The sentence was modified to: The two FBG cross-laying method is adopted to quantitative analysis and get the most sensitive angle α = 30° (the crossover angle between the FBGs is 2α).
- Introduction, please uniform the author names of the literature, e.g. “Wang” and “Guanhua Zhi”.
Response:
Page 2: We have modified these errors in lines 67, 72, 77 and 82. See the revised manuscript submitted for details.
- Line 221, the ranges of the optimized parameters seem a little larger for a simulation result.
Response:
Page 10: The smaller the size of the sensor, the more difficult it is to prepare. In Section 2.1, we simulated ed the effects of silica gel size on the distribution area of the maximum equivalent stress of the proposed sensor. The simulation results show that the proposed sensor size can be adjusted within a certain range according to the thickness and size of the measured object.
The size range the proposed sensor was modified to: a length of 25- 30 mm, width of 8-10 mm and thickness of 2-3 mm.
- Table 8, please add the description on the roles of each FBG.
Response:
Page 16: The table 8 was modified to
Table 8. The detailed parameters of the FBG sensors
|
Parameter |
FBG1 |
FBG2 |
FBG4 |
FBG3 |
||
|
(Bare FBG) |
(Single FBG sensor) |
(dual FBG shape sensor) |
||||
Grating length (mm) |
single mode 1549.190 5 |
single mode 1536.141 5 |
single mode 1529.953 5 |
single mode 1536.185 5 |
||
|
Peak reflectivity (%) |
75 |
75 |
75 |
75 |
||
- Several English errors should be avoided, to name a few
(1) Line 88, “on the one hand”
Response:
Page 2: We have modified this error.
For single-core fiber grating shape sensors, how to eliminate the measurement error caused by distortion to improve the accuracy of shape reconstruction was not considered in the previous literature. In addition, there is little literature on how to eliminate the influence of the properties of soft substrate materials
(2) Line 120, “neutral surface”
Response:
Page 3: We have modified this error in lines 117, 120, 121 and 123

Reviewer 2 Report
This paper reports a dual FBG high-precision shape sensor with high accuracy and reliability in shape reconstruction. I have some comments to do:
- Introduction - I miss a sentence and discussion about the possible use of polymer fiber for such proposed sensor system based on 1 FBG for simulteanous measurement of bending, torsion and strain as reported in: Journal of Lightwave Technology 37 (3), 971-980, 2019. It is very interesting discuss it and refer. https://www.osapublishing.org/jlt/abstract.cfm?uri=jlt-37-3-971
- Figs. 3 to 5 have weak quality. Please improve.
- What are the error associated for the values from all tables in section 3. Nothing is mentioned about it. Please improve.
- Confirm in the Fig. 10 de dimensions of the mold to produce the sensor.
- How about the reproducibility? How many probes were tested? In terms of repeatability is mentioned very well but in terms of reproducibility nothing is achieved.
- Fig. 17: how many probes or FBGs are in? Please be clear.
Author Response
Dear Editor,
Thank you very much for your kind works to our manuscript titled
Design and Measurement of a Dual FBG High-Precision Shape Sensor for Wing Shape Reconstruction
This manuscript was revised carefully according to the comments by the reviewers, so we hope that this revised version is satisfactory.
Best regards!
Sincerely yours,
Huifeng Wu
-----------------------------------------------------------------------
Reviewer: 2
Comments:
This paper reports a dual FBG high-precision shape sensor with high accuracy and reliability in shape reconstruction. I have some comments to do:
- Introduction - I miss a sentence and discussion about the possible use of polymer fiber for such proposed sensor system based on 1 FBG for simulteanous measurement of bending, torsion and strain as reported in: Journal of Lightwave Technology 37 (3), 971-980, 2019. It is very interesting discuss it and refer. https://www.osapublishing.org/jlt/abstract.cfm?uri=jlt-37-3-971
Response:
Page 2: The proposed sensor is a very good idea about this article. We had designed a pressure sensor based on the Fiber Bragg Grating in CYTOP Fiber in our lab.
In addition, we quoted and reviewed this article: “Arnaldo G et al. [27] presented a 3-D displacement sensor based on a single Fiber Bragg Grating in CYTOP Fiber, to obtain the influence of each displacement condition, namely, axial strain, torsion, and bending on the FBG reflection spectrum, and had relative errors below 5.5%, but the Fiber Bragg Grating in CYTOP was prone to breakage and not suitable for harsh environments.”.
- Figs. 3 to 5 have weak quality. Please improve
Response:
Page 5-10 and Page 13: These figures were modified. See the revised manuscript submitted for details.
|
Force + Torque |
- What are the error associated for the values from all tables in section 3. Nothing is mentioned about it. Please improve
Response:
We obtained the data from all tables in section 3 by ANSYS finite element simulation software, The ANSYS finite element simulation software has been developed and applied for many years, the simulation results are very reliable and accurate, so we assume that the error in the simulation results in all tables in section 3 is very small and can be ignored.
- Confirm in the Fig. 10 de dimensions of the mold to produce the sensor.
Response:
Page 12: The figure was modified to
|
1mm |
|
0.3mm |
|
25mm |
Figure 10. Mold of Sensor Fabrication
- How about the reproducibility? How many probes were tested? In terms of repeatability is mentioned very well but in terms of reproducibility nothing is achieved.
Response:
Page 18: For the above repeatability test results, using the same operation procedure, we randomly selected a shape sensor for the test in more detail, two of FBGs in the sensor are labeled as FBG1 and FBG2, respectively.
(a) (b)
Figure 16. Shape sensor FBG error analysis of repeatability:
Page 19: The added content is
(a) (b )
Figure 18. Shape sensor FBG hysteresis error analysis:
- Fig. 17: how many probes or FBGs are in? Please be clear.
Response:
Page19: There are 19 high-precision shape sensors on the test specimen, a total of 38 FBGs in Fig. 17 in the manuscript, we have revised the sentence to:
we will carry out a reconstruction test on a test specimen installed with 19 shape sensors (38 FBGs in total), and the experimental setup is shown in Fig. 17.

Round 2
Reviewer 2 Report
The paper deserves now to be published.